# The role of TAp63γ and P53 point mutations in regulating DNA repair, mutational susceptibility and invasion of bladder cancer cells

Hsiang-Tsui Wang[1†‡], Hyun-Wook Lee[1†], Mao-wen Weng[1†], Yan Liu[2], William C Huang[2], Herbert Lepor[2], Xue-Ru Wu[2], Moon-shong Tang[1*]

[1]Department of Environmental Medicine, New York University Grossman School of Medicine, New York, United States; [2]Department of Urology, New York University Grossman School of Medicine, New York, United States

*For correspondence:
Moon-Shong.Tang@nyumc.org

†These authors contributed equally to this work

Present address: ‡Department of Pharmacology, National Yang Ming Chiao Tung University, Taipei, Taiwan

Competing interest: The authors declare that no competing interests exist.

**Abstract** It has long been recognized that non-muscle-invasive bladder cancer (NMIBC) has a low propensity (20%) of becoming muscle-invasive (MIBC), and that MIBC carry many more p53 point mutations (p53m) than NMIBC (50% vs 10%). MIBC also has a higher mutation burden than NMIBC. These results suggest that DNA repair capacities, mutational susceptibility and p53m are crucial for MIBC development. We found MIBC cells are hypermutable, deficient in DNA repair and have markedly downregulated DNA repair genes, XPC, hOGG1/2 and Ref1, and the tumor suppressor, TAp63γ. In contrast, NMIBC cells are hyperactive in DNA repair and exhibit upregulated DNA repair genes and TAp63γ. A parallel exists in human tumors, as MIBC tissues have markedly lower DNA repair activity, and lower expression of DNA repair genes and TAp63γ compared to NMIBC tissues. Forced TAp63γ expression in MIBC significantly mitigates DNA repair deficiencies and reduces mutational susceptibility. Knockdown of TAp63γ in NMIBC greatly reduces DNA repair capacity and enhances mutational susceptibility. Manipulated TAp63γ expression or knockdown of p53m reduce the invasion of MIBC by 40–60%. However, the combination of p53m knockdown with forced TAp63γ expression reduce the invasion ability to nil suggesting that p53m contributes to invasion phenotype independent from TAp63γ. These results indicate that in BC, TAp63γ regulates DNA repair capacities, mutational susceptibility and invasion, and that p53m contribute to the invasion phenotype. We conclude that concurrent TAp63γ suppression and acquisition of p53m are a major cause for MIBC development.

## Editor's evaluation

This study reveals the connection of p63 and DNA repair in bladder cancer, especially in invasive bladder cancer, which is an understudied area. Results of this study provide new insights into the invasive bladder cancer. This manuscript is of broad interest to the readers in the field of p63, DNA repair and bladder cancer.

## Introduction

Bladder cancer (BC) is the fifth most frequently occurring cancer in the US; in 2018, there were 81,400 new BC cases and 17,980 BC deaths (*Patel et al., 2014*). More than 90 % of BCs are urothelial cell carcinoma and the remaining 10 % are squamous carcinoma, adenocarcinoma, and small cell carcinoma (*Lopez-Beltran and Cheng, 2006*). There are two major forms of BC: the low-grade papillary

types that are well- to moderately-differentiated and do not penetrate the lamina propria of the bladder wall (non-muscle-invasive BC, NMIBC), and the muscle-invasive types (MIBC) that are much more poorly differentiated (*Wu, 2005*; *Van Batavia et al., 2014*). While the former is much less malignant and less lethal than the latter, it frequently recurs after surgery. The molecular paths that lead to these two types of BC are unclear (*Wu, 2005*).

It has long been recognized that most, if not all, tumor cells have the propensity of malignant progression (*Loeb, 2011*). Intriguingly, only 20 % of NMIBC progress into MIBC (*Wu, 2005*; *Knowles and Hurst, 2015*; *Nassar et al., 2019*). Recently, it has been found that high-grade MIBC have a much higher genomic DNA mutation burden than the low-grade NMIBC (*Nassar et al., 2019*; *Pietzak et al., 2017*). These results raise the possibilities that the low frequency of malignant progression of NMIBC is due to these BC cells being hyperactive for DNA repair and less mutable, and that MIBC cells are hypermutable and have a low DNA repair capacity.

To test these possibilities, we determined the susceptibility for DNA-damage-induced mutagenesis and quantified the DNA repair capacity in three well-established MIBC (T24, HT1197 and J82) and two NMIBC (RT4 and RT112/84) cell lines (*Rigby and Franks, 1970*; *Earl et al., 2015*). We found that MIBC cells are more susceptible than NMIBC cells to bulky- and oxidative-DNA-damage-induced mutagenesis, and that although MIBC cells are deficient in nucleotide excision repair (NER) and base excision repair (BER), NMIBC cells are proficient in both repair mechanisms. DNA repair genes such as XPC, Ref1, and hOGG1/2, and TAp63γ, the tumor suppressor isoform of p63, are suppressed in MIBC cells but are highly expressed in NMIBC cells. We found that forced TAp63γ expression up-regulates XPC, Ref1, and hOGG1/2, reduces the mutational susceptibility in MIBC cells, and furthermore reduces the invasion potential of MIBC by about 40–60%. It has been found that more MIBC (50%) carry p53 point mutations (p53m) compared to NMIBC (10%). We found that while knockdown p53m in MIBC reduces invasion potential by 50%, the combination of TAp63γ expression and p53m knockdown converts MIBC to become non-invasive. We also found that T3/T4 late-stage MIBC tissues in humans have lower DNA repair activity and p63 expression than Ta early stage NMIBC tissues. Our results indicate that TAp63γ is a major regulator of DNA excision repair genes that have major effects not only on DNA repair capacities and mutational susceptibility but also on the invasion potential of BC cells. TAp63γ suppression and acquisition of p53m are two major causes for BC becoming invasive.

## Results

### MIBC cells are mutators, whereas NMIBC cells are not

We determined the mutational susceptibility of MIBC and NMIBC cells to DNA-damage-induced mutagenesis in a well-established *supF* forward mutation detection system (*Wang et al., 2013*). First, pSP189 DNA plasmids containing the *supF* gene were either damaged with UVC to induce bulky photodimers, primarily cyclobutane pyrimidine dimers (CPDs), or were modified with $H_2O_2$ to induced oxidative DNA damage (ODD), primarily 8-oxo-deoxyguanosines (8-oxo-dGs) (*Wang et al., 2010*; *Tang, 1996*). These damaged plasmid DNAs were transfected into and replicated in NMIBC cells (RT4), MIBC cells (T24 and HT1197) or immortalized normal urothelial cells (UROtsa). DNA-damage-induced mutations in the *supF* gene in the replicated plasmids were then quantified (*Wang et al., 2010*). The results in *Figure 1A* and *Supplementary file 1* show that UVC-DNA damage induces two-to eightfold more mutations in MIBC cells than in NMIBC cells or UROtsa cells, and that ODD induces six- to eightfold more mutations in MIBC cells compared to NMIBC cells or UROtsa cells. It is worth noting that the susceptibility to DNA-damage-induced mutagenesis of NMIBC cells and UROtsa cells is similar to that of other cultured human cells such as normal lung fibroblasts and skin fibroblasts (*Wang et al., 2010*; *Wang et al., 2012*). These results indicate that compared to NMIBC and normal human cells, MIBC cells are hypermutable.

### MIBC cells are deficient in DNA repair, whereas NMIBC cells are hyperactive in DNA repair

One possible cause that leads to profound differences in susceptibility to UVC- and $H_2O_2$-induced-DNA-damage mutagenesis between MIBC and NMIBC cells is that these cells have different repair capacities. To test this possibility, we determined the capacity of NMIBC (RT4) and MIBC (T24 and HT1197) cells to repair UVC-induced bulky CPDs or $H_2O_2$-induced 8-oxo-dGs in genomic DNA using

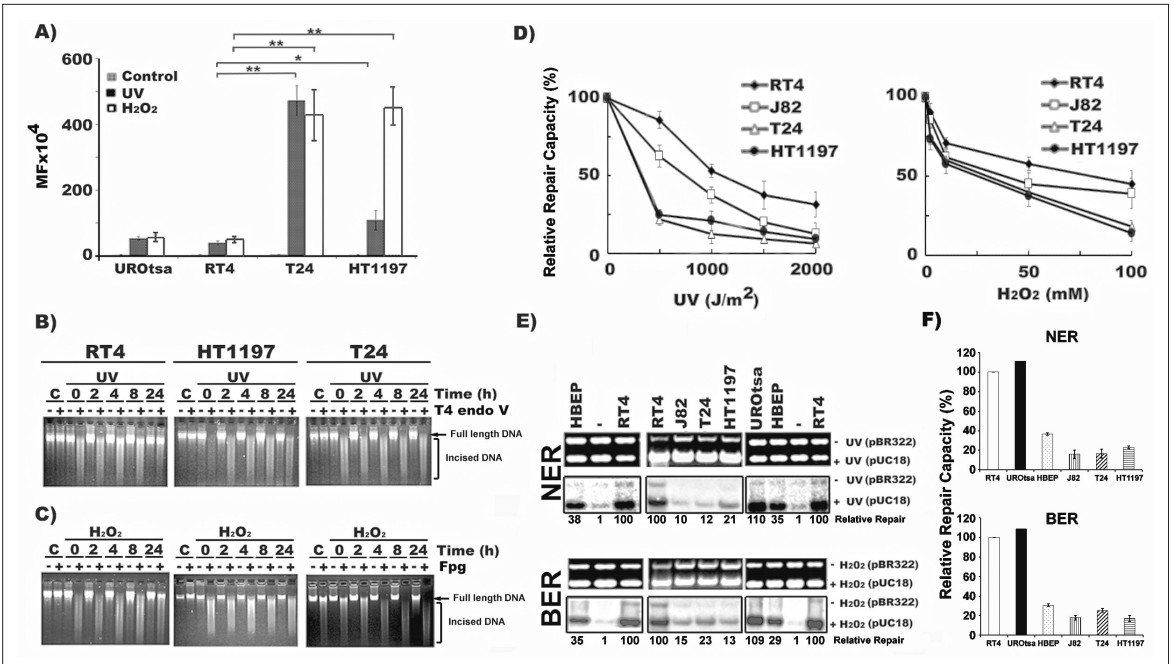

**Figure 1.** MIBC cells are hypermutable and deficient in DNA repair, whereas NMIBC cells are hyperactive in DNA repair. Mutation susceptibility and DNA repair capacity toward UVC- and $H_2O_2$-induced DNA damage were determined in MIBC (T24, HT1197, and J82), NMIBC (RT4), immortalized normal human urothelial (UROtsa) cells and human bladder epithelial progenitor (HBEP) cells, as described (**Wang et al., 2012**). In (**A**), UVC (1500 J/$m^2$)- and $H_2O_2$ (100 mM, 25 °C, 1 hr)-modified pSP189 plasmid DNA containing the *supF* were transfected into different cells and the mutations in the *supF* gene were detected and the mutation frequency (MF) calculated. (**B & C**) Detection of DNA repair in global genomic DNA of NMIBC and MIBC cells. Cells were irradiated with UVC (20 J/$m^2$) or treated with $H_2O_2$ (100 mM, 1 hr, at 37 °C), incubated in growth medium for different times (0, 2, 4, 8, and 24 hr) and the cellular genomic DNAs were isolated. In (**B**), the levels of CPD in the genomic DNA were detected by T4 endonuclease V (T4 Endo V) incision method (**Tang et al., 1994**; **Hu et al., 2002**). In (**C**), the levels of ODD in the genomic DNA were detected by formamidopyrimidine glycosylase (Fpg) incision method (**Wang et al., 2010**). The enzyme-incised resultant DNAs were denatured and separated by electrophoresis, as described (**Wang et al., 2010**). Full length genomic DNA and enzyme-incised DNA are indicated. Symbol: C, Un-irradiated or unmodified control cells. (**D, E and F**) NER and BER capacities in MIBC and NMIBC cells were measured by the host cell reactivation (HCR) assay (**D**), and by the in vitro DNA-damage-dependent repair synthesis (DDDRS) assay (**E & F**) as previously described (**Lee et al., 2014**). For HCR assay, UVC-irradiated (0–2000 J/$m^2$) or $H_2O_2$ (0–100 mM, 25 °C, 1 hr)-modified luciferase reporter (pGL-3-luciferase) and unmodified $\beta$-galactosidase (pSV-$\beta$-galactosidase) plasmids (internal control) were transfected into cells and luciferase and $\beta$-galactosidase activities were measured 72 hr post transfection (**Wang et al., 2012**; **Lee et al., 2014**). The HCR activity was calculated based on the ratio of luciferase activity versus $\beta$-galactosidase activity. The relative repair capacity was calculated based on the luciferase activity obtained from unmodified pGL-3-luciferase activity versus $\beta$-galactosidase activity as 100 %. For DDDRS assay, UVC-irradiated (1500 J/$m^2$) or $H_2O_2$-modified (100 mM, 25 °C, 1 hr) pUC18 and unmodified pBR322 plasmids were used as DNA substrates for DNA repair synthesis carried out in cell extracts isolated from different BC cells (**Lee et al., 2014**). A typical DDDRS assay result is shown in (**E**), in which, the upper panels are ethidium bromide-stained gels. The lower panels are autoradiographs of the same gels. The repair activity was calculated based on the relative intensity of pUC18 band versus pBR322 band. The relative repair capacity depicted at the bottom was calculated based on assigning the repair activity of RT4 cells as 100. (**F**) Quantitation of DDDRS assay results from UROtsa, HBEP, NMIBC (RT4), and MIBC (T24, J82, and HT1197) cells.

The online version of this article includes the following figure supplement(s) for figure 1:

**Figure supplement 1.** NMIBC RT112/84 cells are hyperactive in nucleotide excision repair (NER) and base excision repair (BER).

the T4 endonuclease V (T4 endo V) and formamidopyrimidine glycosylase (Fpg) incision assays, respectively. It has been well-established that T4 endo V specifically incises UV-induced CPDs, and that Fpg incises ODD, mainly 8-oxo-dG adducts (**Latham and Lloyd, 1994**; **Tchou et al., 1994**). Results in **Figure 1B and C** show that, whereas UV-induced CPDs (T4 endo V sensitive sites) and $H_2O_2$-induced ODDs (Fpg sensitive sites) disappear (repaired) within 24 hr incubation in NMIBC cells, these DNA damages remain in MIBC cells. Since it is well established that the major repair pathways for bulky CPDs and ODDs are NER and BER mechanisms, these results indicate that NMIBC cells are proficient in NER and BER, whereas MIBC cells are deficient in NER and BER.

Using the well-established host cell reactivation (HCR) assay and the in vitro DNA–damage-dependent-repair synthesis (DDDRS) assay, we quantified the relative repair capacity in MIBC cells versus NMIBC cells (**Feng et al., 2006**; **Wang et al., 2009**). The results in **Figure 1D** show that the

HCR capacity for UVC-induced DNA damage in MIBC (T24, J82, and HT1197) cells is much lower than in NMIBC (RT4) cells, and that the HCR for $H_2O_2$-induced DNA damage in MIBC cells are also lower than in NMIBC cells although the differences are not as pronounced.

The results in *Figure 1E and F* demonstrate that cell-free lysates isolated from MIBC cells have less than 25 % capacity for repairing UVC- and $H_2O_2$-induced DNA-damages compared to lysates isolated from NMIBC cells. The results also show that cell lysates isolated from NMIBC and UROtsa cells have a higher repair capacity for both UVC- and $H_2O_2$-induced DNA damage than those isolated from normal human bladder epithelial progenitor (HBEP) cells (100 % vs 29–38%), indicating that NMIBC and UROtsa cells have hyperactive NER and BER functions.

## TAp63 is suppressed in MIBC cells and highly expressed in NMIBC cells

It is intriguing that the three MIBC cell lines, T24, HT1197, and J82, which are derived from different patients with T3/T4 MIBC, have similar deficiencies not only in NER but also in BER (*Rigby and Franks, 1970*; *Bubeník et al., 1973*; *O'Toole et al., 1972*). Since multiple repair proteins are involved in NER for UV-induced CPDs, and BER for 8-oxo-dG adducts, and the repair rate-limiting factors such as damage recognition or incision and excision enzymes involved in these two repair pathways do not overlap (*Friedberg Errol et al., 2006*), it is unlikely that these three lines of MIBC cells share the same mutations in genes directly involved in NER and BER. Rather, it is more likely that the deficiencies in both NER and BER in these three MIBC cell lines are due to downregulation of NER and BER genes which are controlled by a common transcription factor and that this transcription factor is abundant in NMIBC but deficient in MIBC. Thus, we inquired what is this gene? In a genome-wide analysis, *Lin et al., 2009* found that p63 controls multiple DNA repair associate genes. Subsequently, *Liu et al., 2012* found that TAp63γ regulates the NER gene XPC in human fibroblasts. It had long been recognized that expression of the p63 gene is suppressed in MIBC cells but not in NMIBC cells *Urist et al., 2002*; these findings led us to determine the role of p63 in DNA repair, mutation susceptibility and invasiveness of BC cells in the current study. The results in *Figure 2A* and *Figure 2—figure supplement 1* show that no p63 proteins, including TAp63 and ΔNp63 isoforms, are non-detectable in the MIBC cells, T24, HT1197 or J82. In contrast, multiple p63 proteins are highly expressed in NMIBC RT4 and HBEP cells. It should be noted that the level of p63 expression in NMIBC cells, is higher than in HBEP and that NMIBC cells have a higher repair activity than HBEP cells (*Figure 1E and F* vs *Figure 2A*).

## TAp63γ regulates NER and BER activity and the mutational susceptibility of BC Cells

More than a dozen of p63 isoforms have been identified and among them TAp63γ, TAp63α, and TAp63β are the three major ones with ΔNp63γ, ΔNp63α, and ΔNp63β as minor forms (*Ghioni et al., 2002*; *Murray-Zmijewski et al., 2006*). We found that no p63 expression, including all the TAp63 and ΔNp63 isoforms, were detected in the MIBC cells such as T24, HT1197, and J82 cells (*Figures 2A and 3* and *Figure 2—figure supplement 1*). On the other hand, the TAp63 isoforms are highly expressed in NMIBC RT4 cells (*Figures 2A and 3* and *Figure 2—figure supplement 1*). These results raise the possibility that it is TAp63 that contribute to the hyperactive DNA repair function in NMIBC cells and that the lack of TAp63 expression may contribute to the DNA repair deficiency in MIBC cells. Two more factors lead us to determine the role of TAp63 isoforms in DNA repair and invasiveness of BC cells. One, TAp63γ has been shown upregulates NER function and expression of repair genes XPC, DDB2 and GADD45 in human skin fibroblasts (*Liu et al., 2012*). While TAp63 isoforms have transcription transactivation function ΔNp63 isoforms lack this function (*Petitjean et al., 2008*). Two, whereas TAp63 isoforms function as tumor suppressors ΔNp63 isoforms have oncogenic function by inhibition of TAp63 function (*Flores, 2007*). It is well established that individuals deficient in DNA repair are more prone to develop cancers (*Friedberg Errol et al., 2006*), and that the somatic cells of these individuals are not only deficient in DNA repair but also more mutagenic (*Friedberg Errol et al., 2006*). These findings lead us to hypothesize the lack of DNA repair and the mutagenic and invasive prone of MIBC is more likely due to lack of TAp63 isoforms rather than lack of the oncogenic ΔNp63 isoforms. Therefore, we next aimed to determine which of these three major isoforms regulates DNA repair and invasion activity in BC cells. To do so, TAp63γ, TAp63α, and TAp63β were each transfected into MIBC T24 cells (*Figure 2B* and *Figure 2—figure supplement 2*) which are deficient in both NER and

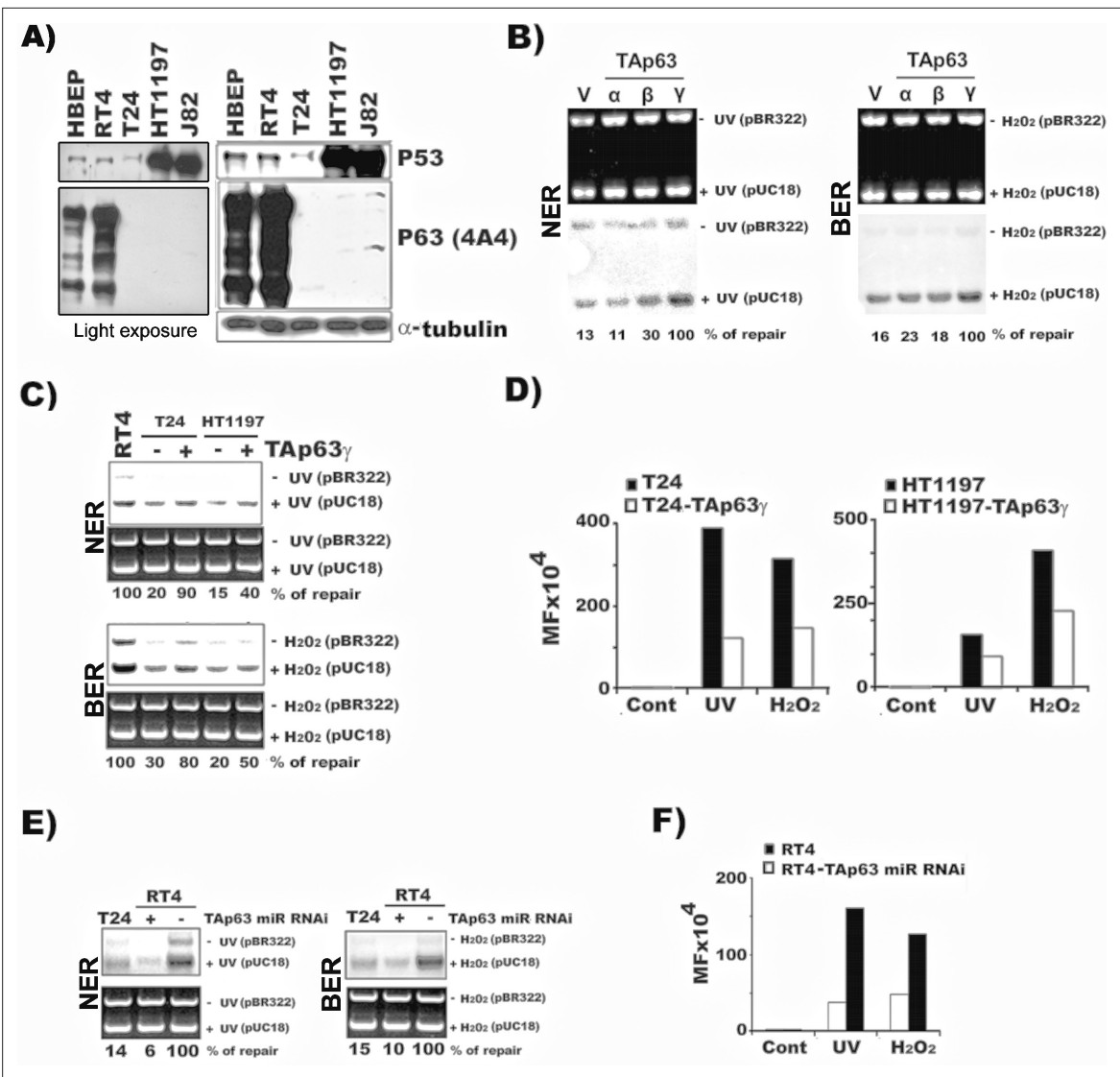

**Figure 2.** TAp63γ regulates NER and BER activity and mutational susceptibility of BC cells. The methods for establishing the stable TAp63α, TAp63β, and TAp63γ in T24 cells (MIBC) and the TAp63γ stable transfectants in HT1197 (MIBC), knockdown TAp63γ by miR-RNAi in RT4 cells (NMIBC), and quantification of NER and BER activity and mutational susceptibility were described in text and in *Figure 1*. (A) Detection of p63 and p53 protein in MIBC (T24, HT1197, J82), NMIBC (RT4) and HBEP cells. Light exposure of p63 gels of RT4 and HBEP shown in the left demonstrates that p63 expression is much higher in RT4 than in HBEP. (B) Detection of NER and BER activity by DDDRS in MIBC T24 cells with and without stable TAp63α, TAp63β, and TAp63γ transfectants. Detection of mRNA of TAp63α, TAp63β, and TAp63γ in T24 cells with and without stable TAp63α, TAp63β, and TAp63γ transfectants was shown in **Figure 3** and *Figure 2—figure supplement 2*. (C & D) Quantification of NER and BER activity by DDDRS (C) and mutational susceptibility (expressed by MF) (D) in T24 and HT1197 cells with and without stable TAp63γ transfectants. (E & F) Quantification of NER and BER activity by DDDRS (E), and mutational susceptibility (F), in NMIBC cells (RT4) with and without stable TAp63γ knockdown by miR-RNAi. Note: (1) stable TAp63α and TAp63β transfects do not affect NER and BER activity significantly in MIBC cells (**B**); (2) stable TAp63γ transfectants enhance NER and BER activity (**B**) and reduce mutational susceptibility (**D**) of MIBC cells, which do not express both TAp63 and ΔNp63 isoforms (*Figure 2—figure supplement 1*); and (3) stable TAp63γ miR-RNAi transfectants reduce NER and BER activity (**E**) and enhance mutational susceptibility (**F**) in NMIBC cells.

The online version of this article includes the following source data and figure supplement(s) for figure 2:

**Source data 1.** Expression of TAp63 and ΔNp63 isoforms in MIBC (T24 and J82), NMIBC (RT4) and immortalized human bladder urothelial UROtsa cells.

**Figure supplement 1.** Expression of TAp63 and ΔNp63 isoforms in MIBC (T24 and J82), NMIBC (RT4) and immortalized human bladder urothelial UROtsa cells.

**Figure supplement 2.** Detection of mRNA of TAp63α and TAp63β in T24 cells with and without stable TAp63α and TAp63β transfectants.

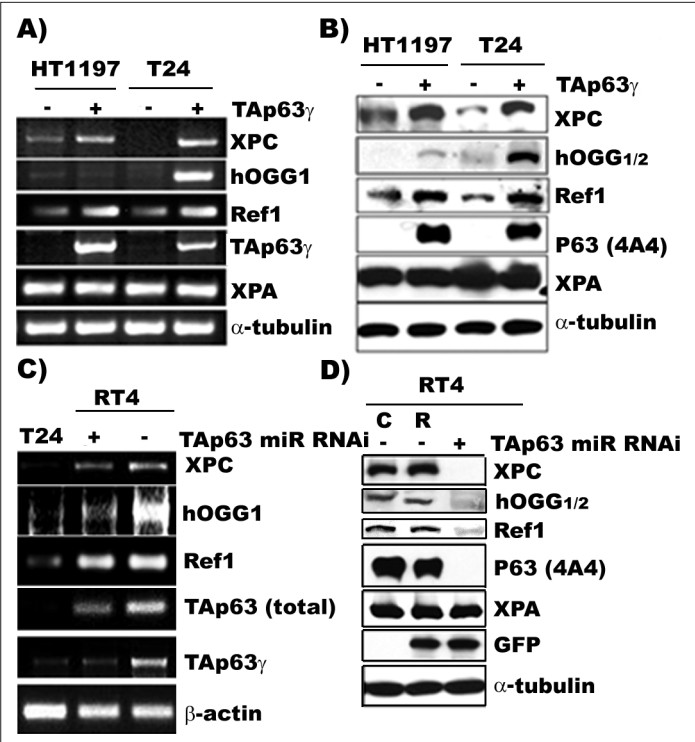

**Figure 3.** TAp63 γ regulates expression of DNA repair genes, XPC, hOGG1/2, and Ref one in BC cells. The constructs of the MIBC (T24 and HT1197) with stable TAp63 γ transfectants and NMIBC (RT4) cells with stable TAp63 γ miR-RNAi are the same as in *Figure 2*. The expression of TAp63 γ , XPA, XPC, hOGG1/2, and Ref1 was determined at the mRNA (**A** & **C**) and protein (**B** & **D**) levels. The methods of detection were the same as described (*Wang et al., 2012*). Note: (1) XPC, hOGG1/2 and Ref1 are downregulated in MIBC cells, T24 and HT1197, and upregulated in NMIBC RT4 cells; (2) forced TAp63 γ expression in MIBC cells upregulates expression of XPC, hOGG1/2, and Ref1 genes; and (3) TAp63 γ knockdown by miR-RNAi downregulates these genes in NMIBC cells. Symbols: C and R in (**D**) represent vector control and vector with random inserts.

The online version of this article includes the following source data and figure supplement(s) for figure 3:

**Source data 1.** Determination of DNA repair capacity and p63 protein levels in normal human urothelial mucosa.

**Source data 2.** Determination of DNA repair capacity and p63 protein levels in normal human urothelial mucosa.

**Figure supplement 1.** Determination of DNA repair capacity and p63 protein levels in normal human urothelial mucosa.

**Figure supplement 2.** The TAp63 promoter region in MIBC cells is rich in transcription silencing markers, H3K9me2 and H3K27me3, whereas this region in NMIBC cells is rich in transcription activating marker H3K4me3.

BER activity as shown in *Figure 1*. The NER and BER capacities in T24 cells with transfected TAp63γ, TAp63α, or TAp63β were determined in comparison to control vector stable transfectants. The results in *Figure 2B* show that T24 cells with stable TAp63γ transfectants have a much higher level of NER and BER activity than T24 cells with stable control vector transfectants. In contrast, the T24 cells with stable TAp63α and TAp63β transfectants have a similar level of NER and BER activity compared to T24 cells with stable control vector transfectants. These results indicate that TAp63γ alone can regulate both NER and BER activity in these BC cells.

We further tested whether there is a causative relationship between TAp63γ expression and NER and BER capacity and susceptibility to DNA damage-induced mutagenesis in NMIBC cells. To do this, the NER and BER capacities and the mutational susceptibilities in two MIBC cells, T24 and HT1197, with and without stable TAp63γ transfectants, were determined. The results in *Figure 2C* show that NER and BER capacity are greatly enhanced in both T24 and HT1197 MIBC cells with TAp63γ stable expression compared to each of the untransfected parental cells. Furthermore, the susceptibility toward UVC- and ODD-induced mutagenesis is also significantly reduced in these MIBC cells with stable TAp63γ expression compared to the parental cells (*Figure 2D* and *Supplementary file 2*).

## Suppression of TAp63γ expression in NMIBC cells reduces DNA repair capacity and enhances mutational susceptibility

To further demonstrate the role of TAp63γ in regulating DNA repair and mutation susceptibility phenotype in BC cells, we determined the DNA repair capacity and susceptibility to DNA damage-induced mutagenesis in NMIBC cells with and without knockdown of TAp63γ. The results in *Figure 2E and F* and *Supplementary file 3* show that knockdown of TAp63γ in the NMIBC cells reduces NER and BER capacities and enhances cell susceptibility to DNA damage induced mutagenesis to the similar level as in MIBC T24 cells. Thus, these results indicate that the upregulation of TAp63γ is a major factor which is responsible for the hyperactive DNA repair function and low susceptibility to DNA-damage-induced mutagenesis in NMIBC cells.

## TAp63γ upregulates XPC, hOGG1/2, and Ref1 in BC cells

Although more than a dozen of different forms of p63 have been identified in human cells, none of them are directly involved in the NER or BER DNA damage processing mechanisms (*Liu et al., 2012*; *Yang et al., 1998*; *Melino et al., 2015*; *Spivak, 2015*; *Marteijn et al., 2014*; *Krokan and Bjørås, 2013*; *Dianov and Hübscher, 2013*). Rather, p63 proteins are transcription factors, which likely regulate NER and BER by affecting the expression of the genes encoding the functional proteins directly involved in NER and BER mechanisms. We tested this possibility, by determining the expression of NER genes, XPA and XPC, as well as BER genes, hOGG1/2, and Ref1 in MIBC (T24 and HT1197) and NMIBC (RT4) cells. XPA is essential for both global genomic and transcription-coupled NER, and XPC is essential for global genomic NER (*Hanawalt and Spivak, 2008*). Ref1 and hOGG1/2 function as repair enzymes for abasic sites and 8-oxo-dG's, respectively (*Xanthoudakis and Curran, 1992*; *Bjorås et al., 1997*). The results in *Figure 3A-D* show that (1) XPC, Ref1, and hOGG1/2 are highly expressed in NMIBC cells but are either not or lowly expressed in MIBC cells; (2) transfection-induced TAp63γ expression elevates levels of XPC, Ref1, and hOGG1/2 in MIBC cells; and (3) knockdown of TAp63γ expression causes a downregulation of XPC, Ref1, and hOGG1/2 in NMIBC cells. Collectively, the results in *Figures 2 and 3* indicate that the level of TAp63γ expression determines the NER and BER activity and the mutational susceptibility phenotype of MIBC and NMIBC cells, and that TAp63γ exerts these effects most likely via its regulation on DNA repair genes such as, XPC, hOGG1/2, and Ref1.

## TAp63γ expression attenuates invasion potential of MIBC cells

Since p63 has been suggested to control tumor invasion and metastasis *via* its regulation of epithelial-mesenchymal transition and expression of matrix metalloproteinase genes (*Celardo et al., 2014*; *Tran et al., 2013*), we examined the effect of the three p63 isoforms: TAp63α, TAp63β, and TAp63γ, on BC migration and invasion using MIBC T24 cells. The results in *Figure 4A* show that compared to parental MIBC T24 cells, the migration and invasiveness of MIBC cells with stable expression of TAp63γ transfectants are greatly reduced (60%). In contrast, MIBC cells with the stable transfectants of TAp63α and TAp63β have the same invasiveness potential as the untransfected parental cells. The stable expression of TAp63γ transfectants in MIBC HT1197 cells also reduced invasion potential (40%) (*Figure 4*). We further tested the effect of TAp63γ on wound healing ability. These results in *Figure 4B* show that MIBC cells, T24 and HT1197, with stable TAp63γ transfectants lose wound healing ability. These results indicate that TAp63γ plays an important role in determining the invasiveness of MIBC.

## TAp63γ expression and P53m knockdown reduce more than 90% of MIBC invasion potential

About 50 % of MIBC have p53 point mutations (p53m) compared to 10 % of NMIBC, and p63 is suppressed in MIBC but is highly expressed in NMIBC (*Figure 2*; *Nassar et al., 2019*; *Urist et al., 2002*; *Wolff et al., 2005*; *Berggren et al., 2001*; *Robertson et al., 2017*; *Cordon-Cardo et al., 1994*; *Spruck et al., 1994*; *Cancer Genome Atlas Research Network, 2014*). We found that HT1197 cells which carry p53m, are 2-fold more invasive than T24 BC cells which carry one wild p53 (p53wt) allele and one p53 allele with a nonsense mutation (*Figure 4A and B*; *Cooper et al., 1994*). While T24 cells have a low level of p53wt, the p53m in HT1197 is highly expressed (*Figure 2A*). These results raise the possibility that p53m may enhance the invasion ability of MIBC independently from TAp63γ. We tested this possibility by determining the effect of knockdown p53m on the invasion ability of MIBC cells. We found that knockdown of p53 expression reduces invasion potential of T24

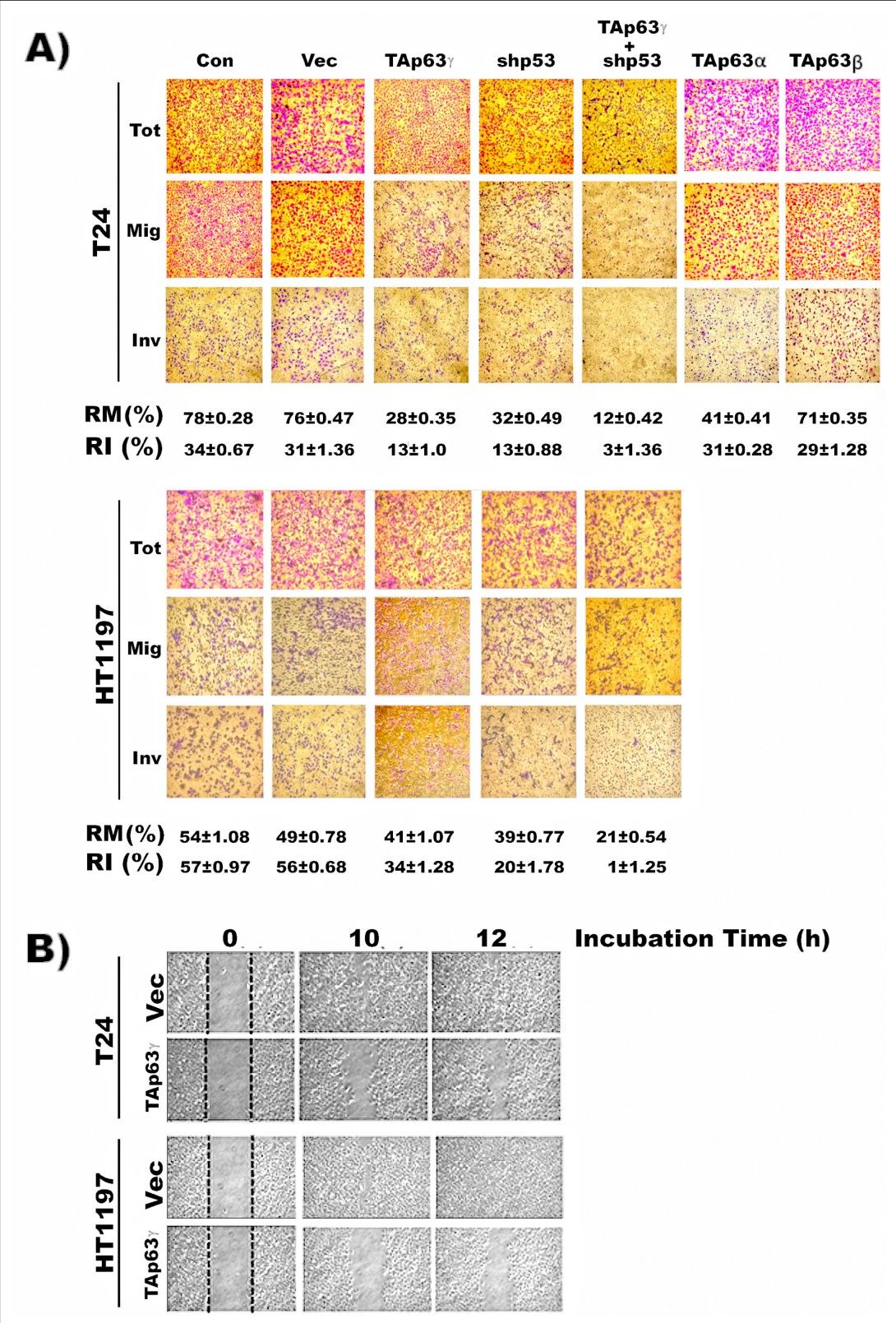

**Figure 4.** TAp63$\gamma$ expression and p53m knockdown reduce invasion potential of MIBC cells. The effect of TAp63$\gamma$ and p53$_m$ on cell mobility and invasion potential were assayed by (**A**) migration and invasion chambers using Matrigel kits, and (**B**) wound healing in different cell lines which were constructed as described in the text. These cell lines are: (1) MIBC (T24 and HT1197) cells, (2) MIBC (T24 and HT1197) cells with stable control vector transfectants, (3) MIBC cells with stable TAp63$\gamma$, TAp63$\alpha$, and TAp63$\beta$ transfectants, (4) MIBC cells with shRNA p53 knockdown, (5) MIBC cells

*Figure 4 continued on next page*

*Figure 4 continued*

with stable TAp63γ transfectants and shRNA p53 knockdown. The relative migration (RM) and relative invasion (RI) potentials were quantified by the methods described in vendor's instruction as follow: $\text{Relative Migration (RM, \%)} = \frac{\text{(Mean of cells migrating through control membrane)}}{\text{(Total cells seeding into the chambers from the beginning)}} \times 100$

$\text{Relative Invasion (RI, \%)} = \frac{\text{(Mean of cells invading through matrigel membrane)}}{\text{(Mean of cells migrating through control membrane)}} \times 100$

Note: Invasion potential of HT1197 cells which over expressed p53$_{m365}$ (*Cooper et al., 1994*) as shown in *Figure 2*, is higher than T24 cells which have a low expression of one wild-type allele and one allele nonsense mutation of p53.

and HT1197 by more than 50 % (*Figure 4A*). Furthermore, the combination of p53 knockdown and forced TAp63γ expression almost completely abolish the invasion potential of MIBC T24 and HT1197 cells (*Figure 4A*). These results indicate that expression of p53m and suppression of TAp63γ are the crucial factors that make these cells invasive.

## DNA repair and P63 expression are inversely related to bladder cancer stage

Although the divergent abilities of cultured MIBC and NMIBC to invade are already well established, their distinct DNA repair capacity and mutation phenotypes, and their relationships with TAp63γ expression presented above have not been reported before. It is possible that some of these characteristics are due to mutations accumulated from generations of in vitro culture growth. Therefore, these characteristics may not be necessarily reflecting the inherent traits of human bladder tumors in vivo. The chance of acquiring "extra" genotypic changes in MIBC cells is even higher than in NMIBC cells since MIBC cells are deficient in DNA repair and are hypermutable. Furthermore, defining the relationship of DNA repair capacity and the level of expression of DNA repair genes and p63 for different grades and stages of human BC is useful for understanding the underlying mechanisms of BC pathogenesis. Therefore, we examined DNA repair capacity and the expression of p63, XPC, Ref1 and hOGG1/2 at the protein level in surgical samples acquired from BC (n = 16) and normal human urothelium mucosa (NHUM) (n = 13). The results show: (1) NHUM, which express UPIIIa (*Lee et al., 2014*), are deficient in NER and BER and in the expression of XPA, XPC, hOGG1/2, Ref1, and p63 genes (*Figures 3 and 5* and *Figure 2—figure supplement 1*). In contrast, cultured human bladder epithelial progenitor (HBEP) cells, which do not express UPIIIa (*Lee et al., 2014*), are proficient in NER and BER and in expressing these NER and BER genes (*Figure 1E*); (2) NER and BER capacity in stage Ta NMIBC is higher than stage T1, T2 MIBC, and stage T3, T4 MIBC have the lowest NER and BER capacity (*Figure 5A,B*); (3) p63 expression is highest in stage Ta NMIBC followed by stage T1 & T2 MIBC, and stage T3 & T4 MIBC have lowest p63 expression (*Figure 5C and D*); and (4) stage T3 & T4 MIBC have lower XPC, XPA,

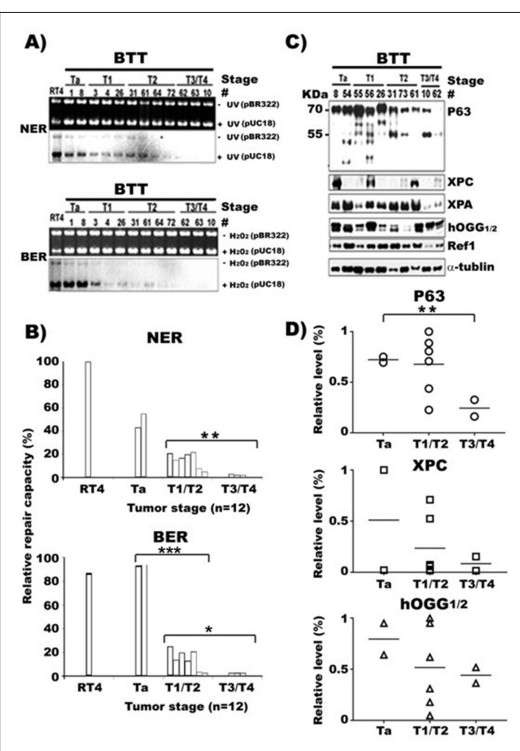

**Figure 5.** NER and BER capacity, expression of repair genes, and p63 in human bladder cancer tissues of different stages. Cell lysates were prepared from human bladder tumor tissues (BTT) of different stages (Ta to T4, n = 16). (**A**) NER and BER capacity in these lysates were determined by in vitro DNA-damage-dependent-repair synthesis assay, as described in *Figures 1 and 2*, and relative repair capacities were calculated and depicted in (**B**). (**C**) Repair proteins XPC, hOGG1/2 and Ref1, and p63 proteins in these lysates were detected the same as described in *Figure 3*. (**D**) Quantifications of the relative levels of p63, XPC, hOGG1/2, and Ref1 proteins in different stage BC tissues. * and *** represent p value of < 0.05 and < 0.001. The p value for the NER difference of Ta vs T1/T2 in (**B**) is 0.073.

The online version of this article includes the following source data for figure 5:

**Source data 1.** Detection of P63 and diferent repair proteins in human bladder cancer tissues.

and Ref1 expression than Stage Ta to T2 MIBC (*Figure 5C and D*). These results demonstrate that stage Ta NMIBC and stage T3 & T4 MIBC have similar NER and BER capacity to that in cultured NMIBC and MIBC cells, respectively. In contrast, NER and BER capacity in stage T1 & T2 MIBC are lower than Ta NMIBC but higher than T3 & T4 MIBC. These results demonstrate that both cultured MIBC cells and late stage MIBC tissues are deficient in NER and BER and p63 expression, while Ta NMIBC tissues and cultured NMIBC cells have relatively high p63 expression and high NER and BER capacity. In contrast, stage T1 and T2 MIBC tissues have intermediate levels of NER and BER capacity and p63 expression. These results indicate that expression of p63 and DNA repair capacity play a pivotal role in BC pathogenesis.

## Discussion

We found that the well-established NMIBC cells, RT4, and two MIBC cell lines, T24, and HT1197, have distinct differences in their DNA repair capacity, expression of DNA repair genes, hOGG1/2, Ref1, and XPC, and expression of the tumor suppressor gene TAp63γ, as well as marked differences in their susceptibilities to DNA-damage-induced mutagenesis. The two MIBC lines T24 and HT1197, have similar, if not identical, deficiencies in NER and BER, as well as low expression of hOGG1/2, Ref1, XPC, and p63 genes, and both exhibit a hypermutable phenotype. Since these two cell lines were derived from different patients with T3 and T4 stage MIBC (*Rigby and Franks, 1970*; *O'Toole et al., 1972*; *Rasheed et al., 1977*), it is unlikely these molecular features were obtained during prolonged in vitro culture conditions. Instead, it is more likely that they are the molecular traits of human MIBC. It should be noted that a third MIBC cell line, J82, also exhibits deficient p63 expression, and deficiency in NER and BER as with T24 and HT1197 cells (*Figures 1D, E, F , and 2A*), and that a second NMIBC RT112/84 cell line has high DNA repair capacity similar to that found in RT4 cells (*Figure 1—figure supplement 1*). We found that while stage Ta NMIBC tissues are proficient in NER and BER, stage T3 and T4 MIBC tissues are deficient in both. Based on these results we reached the following six conclusions: (1) MIBC cells are hypermutable, whereas NMIBC cells are not; (2) MIBC cells are deficient in NER and BER, whereas NMIBC cells are hyperactive in NER and BER; (3) the hypermutable phenotype of MIBC cells is the result of a deficiency in DNA repair; (4) TAp63γ is suppressed in MIBC cells and highly expressed in NMIBC cells; (5) TAp63γ upregulates XPC, Ref1, hOGG1/2, and possibly other repair genes, and consequently regulates the NER and BER activity; BC cells with down-regulated TAp63γ are deficient in NER and BER and are hypermutable; and (6) TAp63γ plays a major role in MIBC development *via* both DNA-repair-activity-dependent and independent mechanisms.

It is well established that XPC is essential for repair of bulky DNA damage, and that hOGG1/2 and Ref1 are the major enzymes for repair of 8-oxo-dGs and abasic sites in genomic DNA, therefore suppression of these three DNA repair genes is sufficient to enhance both bulky DNA adducts and ODD-induced mutagenesis (*David et al., 2007*). We found that the expression of TAp63γ in MIBC not only restores the DNA repair capacity but also greatly reduces DNA-damage-induced mutations (*Figure 2C and D*). Furthermore, suppression of TAp63γ in NMIBC cells not only reduces DNA repair capacity, it also greatly enhances DNA-damage-induced mutations (*Figure 2E and F*). These results lead us to conclude that the hypermutable phenotype of MIBC cells is the result of a deficiency in DNA repair of these cells, which is due to suppression of TAp63γ, and that the non-mutator phenotype of NMIBC cells is due to the hyperactive DNA repair functions.

Our results show that XPA is expressed in both NMIBC (RT4) and MIBC (T24 and HT1197) cells and that neither forced TAp63γ expression in MIBC cells nor knockdown TAp63g expression in NMIBC cells affects the XPA expression in these cells (*Figure 2*). These results indicate that XPA expression is not regulated by TAp63γ. Similar results were also reported by *Liu et al., 2012*. However, our results show that the levels of both XPA and TAp63 in bladder tumors isolated from T3/T4 patients are relatively low compared to those in T1/T2 and Ta patients. These results raise the possibility that XPA was downregulated at the late stage of MIBC development, however, the mechanism is unclear. p63 expression is essential for the normal development of bladder tissue (*Cheng et al., 2006*; *Yang et al., 1999*; *Karni-Schmidt et al., 2011*). In mouse models it has been found that p63 knockout is developmentally lethal and that only one layer of bladder tissue is developed in these mice (*Yang et al., 1999*). In the human bladder, p63 is expressed in three layers of urothelial cells but not in the outer layer of urothelial cells (*Ho et al., 2012*). Consistent with these results, we found that p63 is not expressed in normal human urothelial mucosa (NHUM) which also is deficient in DNA repair

(*Figure 3—figure supplement 1*). However, we found that most BC tumor tissues we examined show some degree of p63 expression, with T3/T4 BC tissues expressing less than T1/T2 BC tissues and Ta BC tissues expressing higher p63 than T1/T2 BC tissues. The p63 expression in these BC tissues shows a positive relationship with DNA repair activity in that Ta BC tissues have higher repair capacities than T1/T2 BC tissues and T3/T4 BC tissues have lowest DNA repair capacity (*Figure 5*). These results raise an intriguing question: which layers of urothelial cells give rise to bladder tumors, the outer layer of urothelial cells, which lack p63 expression and DNA repair activity or the basal layer urothelial cells, which have both? We propose that NMIBC is derived from urothelial cells which have TAp63γ expression and are proficient in DNA repair and that MIBC is derived from urothelial cells which have TAp63γ suppressed and a lack of DNA repair (*Van Batavia et al., 2014*).

Our results also raise the additional important questions regarding what mechanisms control TAp63γ expression in BC cells and how do bladder carcinogens affect these mechanisms? We found that expression of TAp63γ in MIBC and NMIBC cells is associated with H3K4me3, H3K9me2, and H3K27me3 markers (*Figure 1* and *Figure 3—figure supplement 2*). It has been found that the p63 gene is rarely mutated in human cancer (*Cancer Genome Atlas Research Network, 2014*). These results suggest the possibility that TAp63γ expression is regulated epigenetically rather than by mutations, and that suppression of TAp63γ in BC cells is a gradual process rather than an all-or-none process (*Figure 5C and D*). It also suggests that the level of TAp63γ expression determines the progression of this disease and that the late-stage BC cells are more mutable than the early-stage BC cells.

It has been well established that only 20 % of NMIBC develop into MIBC. While 50 % of the MIBC have p53 point mutations, NMIBC rarely have p53 point mutations (*Nassar et al., 2019*; *Robertson et al., 2017*; *Cordon-Cardo et al., 1994*; *Spruck et al., 1994*; *Cancer Genome Atlas Research Network, 2014*). These results indicate that p53 point mutations are crucial for MIBC development. We propose that the over expression of the p63 gene and hyperactive DNA repair in NMIBC cells may prevent the acquisition of p53 point mutations, thereby inhibiting the development of MIBC.

We found that TAp63γ expression can restore DNA repair capacity and reduce DNA damage induced mutagenesis to levels equal to the levels in wild-type cells (*Figures 2 and 3*). However, TAp63γ overexpression or knockdown p53m alone in the MIBC cells cannot fully reduce the cell invasion to the level of NMIBC cells (*Figure 4*). These results indicate that the invasion phenotypes are controlled by multiple mechanisms and that the p63 pathway is two of them. If this is the case, then MIBC with stable TAp63γ transfectants and NMIBC cells with suppressed TAp63γ and stable p53m transfectants will be able to provide us with well-characterized research platforms to search the additional mechanisms for cell invasion other than those that involve the p63 and p53 pathways.

Finally, the results showing that T1/T2 and T3/T4 BC cells are deficient in NER and BER, provide a strong rationale for exploiting DNA damaging agents as potential therapeutic agents for MIBC.

## Materials and methods
### Materials
XPA, XPC, hOGG1/2, p53, and p63 antibodies were purchased from Santa Cruz Biotechnology (Dallas, TX), α-tubulin from Calbiochem (Billerica, MA), IgG antibodies from Amersham Biosciences (Pittsburgh, PA), T4 kinase, protease K, and RNase A from New England Biolabs (Ipswich, MA), α-$^{32}$P-dATP from Perkin Elmer (Waltham, MA), and growth medium MEM and EMEM from Sigma and ATCC, respectively. Plasmids were prepared, as described (*Wang et al., 2013*).

### Normal human urothelial mucosa and bladder tumor tissues
Normal human urothelial mucosa was obtained by excisional biopsy (n = 13) as previously described (*Bouaoun et al., 2016*). Bladder tumors of various grades and stages (n = 16) were obtained by surgical resection, and the tissue samples were immediately frozen at –80 °C.

### Cells and cell culture
RT4, T24, J82, and HT1197 cells were purchased from ATCC, UROtsa from Thermo Fisher Scientific, RT112/84 from Sigma-Aldrich, and HBEP from CELLnTEC (Switzerland). All these cells and cell lines were validated to be free of mycoplasma and were authenticated by the vendors by short tandem repeat profiling at the time of purchase. All the cell lines were used within 6 months of purchase.

HBEP cells were cultured in the CnT58 medium (CELLnTEC, Switzerland). UROtsa cells were grown as described (*Rossi et al., 2001*). Human bladder tumor cells, RT4 and T24, were grown in McCoy's 5 A supplemented with 10 % FBS; HT1197 and J82 were grown in MEM supplemented with 10 % FBS; RT112/84 cells were grown in EMEM supplemented with 2 mM glutamine, 1 % non-essential amino acids and 10 % FBS.

## Constructions of BC cells with different stable transfectants

Plasmid DNA constructs with CMV promoter controlled TAp63γ, TAp63α, and TAp63β were used to construct stable transfectants in BC cells (*Liu et al., 2012*). Stable TAp63γ and p53 knockdown BC cells were constructed using microRNA (for TAp63γ) and short hairpin (sh)RNA (for p53) (*Chang, 2004*).

## DNA repair assays

Germicidal lamp with low pressure mercury which emits 254 nm (95%) as UVC source for irradiation of plasmid DNA and cultured human cells. T4 endonuclease V (T4 endo V) incision assay for UVC-induced CPD in genomic DNA and formamidopyrimidine glycosylase (Fpg) incision assay for 8-oxo-dG are the same as previously described (*Wang et al., 2010*; *Arakawa et al., 2012*). Host cell reactivation (HCR) and DNA-damage-dependent-repair synthesis were determined as described (*Wang et al., 2013*; *Tang, 1996*; *Lee et al., 2014*; *Lee et al., 2018*).

## The *supF* mutation assay

DNA-damage-induced mutations in the *supF* gene in pSP189 plasmid were detected the same as previously described (*Wang et al., 2013*; *Wang et al., 2012*; *Wang et al., 2009*).

## Reverse transcription-polymerase chain reaction and western blot assay

The methods for detection of TAp63γ TAp63β,TAp63α, XPA, XPC, hOGG1/2, and Ref1 at mRNA and protein levels were as previously described (*Wang et al., 2012*).

## Cell motility assay

BC cell migration ability and invasion potential were assessed by using a Matrigel Invasion Chamber with Control Inserts and Matrigel Inserts following the vendor's instruction (BD BioCoat Matrigel Invasion Chamber, Fisher Scientific). Cell motility activity was also examined by scratch-wound assay (*Liang et al., 2007*).

## Chromatin immunoprecipitation (ChIP)

The ChIP assays were performed using EZ ChIP Chromatin Immunoprecipitation Kit (EMD Millipore, Darmstadt, Germany, 17–371) according to manufacturers' instructions. Primers (5'–3') were TTTTAG-CCTCCCGGCTTTAT and ACGAAACGCTGGATGTAAGG for TAp63.

## Study approval

The acquisition of tissue from patients was reviewed and approved by the Institutional Review Board (IRB, #H10181) of the New York University Langone Medical Center and the School of Medicine BioRepository Center. Informed consents were for research and HIPAA authorization were obtained from patients for tissue collections.

## Acknowledgements

We thank Drs. Frederic Beland and Catherine B Klein for reviewing this manuscript. This work was supported by National Institutes of Health grants [R01CA190678, 1P01CA165980, P30CA16087 and ES00260].

## Additional information

### Funding

| Funder | Grant reference number | Author |
|---|---|---|
| National Cancer Institute | P30CA16087 | Moon-shong Tang |
| National Institute of Environmental Health Sciences | ES00260 | Moon-shong Tang |
| National Cancer Institute | R01CA190678 | Moon-shong Tang |
| National Cancer Institute | Cancer Etiology 1P01CA165980 | Moon-shong Tang |

The funders had no role in study design, data collection and interpretation, or the decision to submit the work for publication.

### Author contributions

Hsiang-Tsui Wang, Mao-wen Weng, Conceptualization, Data curation, Formal analysis, Investigation, Methodology, Validation, Visualization, Writing – original draft, Writing – review and editing; Hyun-Wook Lee, Conceptualization, Data curation, Investigation, Methodology, Validation, Visualization, Writing – original draft, Writing – review and editing; Yan Liu, Investigation, Methodology, Validation, Writing – original draft; William C Huang, Data curation, Investigation, Methodology, Resources, Validation, Writing – original draft; Herbert Lepor, Funding acquisition, Investigation, Methodology, Resources, Validation, Writing – original draft; Xue-Ru Wu, Conceptualization, Funding acquisition, Investigation, Methodology, Resources, Validation, Writing – original draft; Moon-shong Tang, Conceptualization, Data curation, Formal analysis, Funding acquisition, Investigation, Methodology, Project administration, Resources, Supervision, Validation, Visualization, Writing – original draft, Writing – review and editing

### Author ORCIDs

Moon-shong Tang http://orcid.org/0000-0001-7898-6322

### Decision letter and Author response

Decision letter https://doi.org/10.7554/eLife.71184.sa1
Author response https://doi.org/10.7554/eLife.71184.sa2

## Additional files

### Supplementary files

• Supplementary file 1. UV- and $H_2O_2$-DNA damage induce more mutations in MIBC (T24 & HT1197) than NMIBC (RT4) cells and normal human urothelial (UROtsa) cells.

• Supplementary file 2. Enforced TAp63γ expression reduces UV- and $H_2O_2$-DNA damage[2] induced mutations[3] in MIBC (T24 & HT1197) cells.

• Supplementary file 3. Knockdown TAp63γ expression enhances UV- and $H_2O_2$-DNA damage induced mutations in NMIBC (RT4) cells.

• Transparent reporting form

### Data availability

All data generated or analysed during this study are included in the manuscript and supporting file; Source Data files have been provided for Figures 1 to 5.

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
