## [Editor Report]

This study reveals the connection of p63 and DNA repair in bladder cancer, especially in invasive bladder cancer, which is an understudied area. Results of this study provide new insights into the invasive bladder cancer. This manuscript is of broad interest to the readers in the field of p63, DNA repair and bladder cancer.

---

## [Decision Letter]

**Decision letter after peer review:**

Thank you for submitting your article "The role of TAp63g and p53 point mutations in regulating DNA repair, mutational susceptibility and invasion of bladder cancer cells" for consideration by *eLife*. Your article has been reviewed by 3 peer reviewers, one of whom is a member of our Board of Reviewing Editors, and the evaluation has been overseen by Maureen Murphy as the Senior Editor. The following individual involved in review of your submission has agreed to reveal their identity: Darel J Hunting (Reviewer #3).

Essential revisions:

1) The role and mechanism(s) of selective mutant p53 in migration and invasion in bladder cancer are characterized in this study, but the reviewers feel that the data presented do not fully support the conclusions. More specifically: two invasive bladder cancer cell lines studied here contain p53 mutations outside the DNA binding domain, and it is unclear whether these 2 mutant p53 proteins have GOF activity. It is also unclear whether the observation made from these 2 cell lines accurately reflects the role of mutant p53 in bladder cancer in general. Among the 3 hotspot mutant p53 (175, 248 and 273), only 248 exhibits a promoting effect on migration and invasion of bladder cancer cells, and this was tested only in 1 cell line with ectopic overexpression. The authors should provide additional experimental evidence with true GOF mutant alleles of p53.

2) The authors should examine the regulation of DNA repair by δ-N forms in bladder cancer cells or explain the rationale for examining TA isoforms only in this study.

3) The authors examined the effect of ectopic TAp63γ expression in invasive bladder cancer only (not in non-invasive bladder cancer) and TAp63 knockdown in non-invasive bladder cancer only (not in invasive bladder cancer). It is requested that the authors perform these experiments in both type of cells to compare whether there are difference between invasive and non-invasive bladder cancer cells or provide rationale for the current experiment design.

4) Please provide quantitation for migration, invasion, and wound healing experiments to increase the clarity, and provide error bars for the quantitative results.

5) The statement 'It should be noted that the level of p63 expression in NMIBC cells, is higher than in HBEP and that NMIBC cells have a higher repair activity than HBEP cells.' (line 164-165) is not supported by Figure 2A; please clarify.

6) Figure 3 shows that TAp63γ does not change XPA expression in bladder cancers. Therefore, the association of XPA and TAp63γ in bladder patient samples (Figure S3) may not be due to the direct regulation of XPA by TAp63. This should be addressed in the Discussion.

7) Please discuss and clarify the virtual absence of both NER and BER in the NMIBC cells. Is it because these cells are sickly and difficult to culture? These tumors should be extremely sensitive to DNA damaging agent, such as ionizing radiation.

8) The high levels of unmutated (WT) p53 expression in NMIBC cells should lead to cell cycle arrest and possibly apoptosis unless a downstream protein is mutated or altered in expression. It is requested that the authors test whether the p53 damage response pathway is functional in these cells.

9) Please discuss the potential role of low levels p63 in MIBC in double strand break repair.

10) The authors state that the NMIBC cells are hyperactive for DNA repair but they seem to be normal. Please tone down this conclusion or clarify it with data or examples from the literature.

11) Please explain why the two wavelengths of UV are used in the two assay in Figure 1.

*Reviewer #1:*

Figure 2A shows that all types of p63 isoforms are largely disappeared in invasive bladder cancer lines, indicating the down regulation of both TA and δ-N forms. It is suggested to examine the regulation of DNA repair by δ-N forms in these cells.

Authors examined the effect of ectopic TAp63γ expression in invasive bladder cancer only (not in non-invasive bladder cancer) and TAp63 knockdown in non-invasive bladder cancer only (not in invasive bladder cancer). It is suggested to perform these experiments in both type of cells to compare whether there are difference between invasive and non-invasive bladder cancer cells.

Figure 5 shows that in bladder cancer cell lines tested, only selective mutant p53 proteins exhibit the promoting effects on migration and invasion. The role of mutant p53 in migration and invasion in bladder cancer needs to be further characterized. Since the main topic of this study is the repair capacity in bladder cancer cell lines and the connection between p63 and DNA repair in bladder cancer cells, authors may consider remove this part of the study.

Please provide quantitation for migration, invasion, and wound healing experiments to increase the clarity, and provide error bars for the quantitative results.

*Reviewer #2:*

In Figure 2A, the weak exposure of p63 (4A4) blot should be presented, in order to observe the expression difference of p63 between HBEP and RT4 cells.

In Figure 3A, 3B and 3D, the authors should remove the XPA blots, otherwise, the authors would like to discuss in the main text.

I suggest to remove the Figure 4A and Figure 5A. This study lacks mechanical analyses (DNA repair, gene expression etc.) to support the idea that mutant p53 plays a role here. In addition, the data from cell lines used in this study cannot support the main statement for these two figures.

*Reviewer #3:*

There are however three points which should be addressed in the Discussion: the virtual absence of both NER and BER in the NMIBC cells bothers me. I would expect these cells to be sickly and difficult to culture. In addition, these tumors should be extremely sensitive to DNA damaging agent, such as ionizing radiation. Secondly, the high levels of unmutated p53 expression in NMIBC cells should lead to cell cycle arrest and possibly apoptosis unless a downstream protein is mutated or underexpressed. It would have been good to know if the p53 damage response pathway was functional in these cells. However, this information is not essential to support the conclusions of this study.

Finally, many cancer cell types have defects in double strand break repair. The low levels of p53 in MIBC cells should affect DSB repair. The authors could discuss this.

---

## [Author Response]

Essential revisions:1) The role and mechanism(s) of selective mutant p53 in migration and invasion in bladder cancer are characterized in this study, but the reviewers feel that the data presented do not fully support the conclusions. More specifically: two invasive bladder cancer cell lines studied here contain p53 mutations outside the DNA binding domain, and it is unclear whether these 2 mutant p53 proteins have GOF activity. It is also unclear whether the observation made from these 2 cell lines accurately reflects the role of mutant p53 in bladder cancer in general. Among the 3 hotspot mutant p53 (175, 248 and 273), only 248 exhibits a promoting effect on migration and invasion of bladder cancer cells, and this was tested only in 1 cell line with ectopic overexpression. The authors should provide additional experimental evidence with true GOF mutant alleles of p53.

We agreed with reviewer’s critique that results presented in Figure 5 are lacking an assessment of the mechanistic role of p53 point mutation in BC invasion development, and that we do not have a convincing explanation of why p53_m248_ and p53_m273_ but not p53_m175_ can promote the invasive potential of the non-invasive bladder cancer cells RT4. More experiments are needed to propose that these mutant p53 proteins have “gain-of-function” in bladder cancer cells, which may take a major effort to achieve this goal. Therefore, we removed the results in Figure 5 and eliminated all statements on the “gain-of function” as reviewer suggested.

Having said that we believe our results show direct relationship between p53 point mutations and bladder cancer invasion ability. Our results show that knockdown p53 in invasive BC cells, J82, T24 and HT1197 which carry p53 mutations in both alleles (J82, p53_m271_/p53_m271_), and one allele (T24, p53_m126_/in flame deletion, HT1197, p53m365/wt) reduces both migration and the invasion ability to 50%. These results indicate that factors other than p53m contribute to the invasion ability of these cells. It is well established that 50% of MIBC carry p53 point mutations with codons 248, 273, 280, 282 and 285 as mutation hotspots. In contrast, less than 10% of non-invasive bladder cancers carry p53 mutation. Since all three MIBC cells lines bear one or two alleles with p53 point mutation these results strongly suggest that mutated p53 proteins play an important role in the invasive property of these cancer cells. Our results shown in Figure 4 that knockdown of the p53 expression in T24, J82 and HT1197 can further reduce the invasiveness of these cancer cells significantly. The results from J82 cells are particularly strong, since both p53 alleles in J82 cells are mutated, and knockdown of the expression of the p53m271 reduces the invasion potential by 50%. We believe these results support the clinical findings that the mutated p53 protein may play important role in making bladder cancer invasive.

2) The authors should examine the regulation of DNA repair by δ-N forms in bladder cancer cells or explain the rationale for examining TA isoforms only in this study.

There are two major reasons for examining the roles of TAp63 isoforms but not ΔNp63 isoforms in regulating DNA repair and cell invasion function in bladder cancer cells. First, we have found that no p63 expression, including all the TAp63 and ΔNp63 isoforms, at either the protein and mRNA level, were detected in the muscle invasive bladder cancer (MIBC) cells such as T24, HT1197 and J82 cells (Figure 2 and Supplementary S-Figure 2). On the other hand, the TAp isoforms are highly expressed in non-invasive bladder cells (NMIBC) RT4 at both mRNA and protein level (Figure 2 and Supplementary S-Figure 2). At protein level ΔNp63 isoforms were barely or not detectable in the NMIBC RT4 cells (Figure 2 and Supplementary S-Figure 2). It has been established that TAp63 isoforms have transcription activation function while ΔNp63 isoforms lack this function. Since we found that MIBC cells are deficient in NER and BER and have a suppressed expression of not only multiple DNA repair genes, such as XPC, hOGG1/2 and Ref1, but also TAP63γ. In contrast, both DNA repair genes and TAp63γ are highly expressed in NMIBC cells. These results raise the possibility the expression of these multiple repair genes requires transcription activation of TAp63γ. Second, it has been found that ΔNp63 isoforms do not function as transcription activation factor (lack of transcription activation domain) (Petitjean et al., carcinogenesis, 2008). It has been suggested that DNp63 isoforms function as inhibitors for TAp63 isoforms (via polymerization with monomeric TAp63?). Since at protein level ΔNp63 isoforms were not detectable in the MIBC, and were barely detectable or not detectable in NMIBC, these results strongly suggest that the deficient of DNA repair of MIBC has nothing to do with ΔNp63 function, and ΔNp63 isoforms may not play important role in both DNA repair and tumor invasion function.

Our results confirm that both of these reasons are correct. Our results are both necessary and sufficient to demonstrate that TAp63g role in DNA repair and invasion. These are the reasons we focused our research on determining the function of TAp63. We believe the same rationales are shared by many researchers in this area. For example, Liu et al., found that TAp63γ upregulates NER (DNA repair, 2012).

There are more than six TAp63 isoforms and six ΔNp63 isoforms have been identified in human cells. The results that MIBC cells are lacking expression of TAp63g and multiple DNA repair genes, and that while NMIBC cells lack ΔNp63 expression, they have abundant expression of TAp63γ and DNA repair genes, and that TAp63 isoforms have transcription activation function while ΔNp63 isoforms do not has led us to make a reasonable practical decision to test the role of major TAp63 isoforms including TAp63g rather than testing DNp63 isoforms in DNA repair function.

3) The authors examined the effect of ectopic TAp63γ expression in invasive bladder cancer only (not in non-invasive bladder cancer) and TAp63 knockdown in non-invasive bladder cancer only (not in invasive bladder cancer). It is requested that the authors perform these experiments in both type of cells to compare whether there are difference between invasive and non-invasive bladder cancer cells or provide rationale for the current experiment design.

Our results show that MIBC cells are deficient in TAp63γ expression and repair, in contrast, NMIBC cells are repair proficient and have normal to hyper-expression of TAp63γ. To further find out the causative role of the expression of TAP63γ and the repair capacity in these two types of cells, we determined whether or not changing the expression of this isoform in MIBC and NMIBC would result in changing the repair capacity and invasion phenotypes. Our results in Figures 2-4 demonstrate that it is indeed the case. Namely, the TAp63γ expression is clearly related to the DNA repair capacity and invasion phenotypes of NMIBC and MIBC. The biochemistry results also clearly support the idea that TAp63γ regulates the DNA repair function (Figure 3). We believe these results are necessary and sufficient to support the hypothesis that TAp63γ regulates both repair and invasion function.

Since MIBC cells are deficient in TAp63γ already, it is difficult to envision that further knocking down this gene expression (to reduce p63 expression in the cells which have zero p63 expression) will provide more information on the role of this gene in DNA repair and invasion. The same is true in the case of NMIBC, since TAp63γ is highly expressed in these cells and these cells are repair proficient and non-invasive, it is difficult to envision further forced TAp63γ expression would yield more information on the role repair and invasion. Therefore, we believe that further knocking down the TAP63γ in cells which are deficient in the expression of this gene, and forced expression of this gene in cells which are already proficient in its expression, will not render more information on the role of this protein in DNA repair and invasion function.

4) Please provide quantitation for migration, invasion, and wound healing experiments to increase the clarity, and provide error bars for the quantitative results.

We added the quantification results with error bars on migration in addition to the invasion results in Figure 4 in the revised manuscript. The wound healing assay is a different approach to demonstrate the effect of TAp63γ on the migration ability of MIBC and NMIBC. We believe this method at best is semiquantitative. Furthermore, the results are consistent with the results obtained from the chamber migration method in which we have presented the proper quantified results.

5) The statement 'It should be noted that the level of p63 expression in NMIBC cells, is higher than in HBEP and that NMIBC cells have a higher repair activity than HBEP cells.' (line 164-165) is not supported by Figure 2A; please clarify.

In the revised manuscript, we presented the results which show that the expression of p63 in NMIBC is higher than in HBEP (Figure 2A) and that NMIBC cells have a higher repair activity than HBEP cells (Figure 1 E and 1 F).

6) Figure 3 shows that TAp63γ does not change XPA expression in bladder cancers. Therefore, the association of XPA and TAp63γ in bladder patient samples (Figure S3) may not be due to the direct regulation of XPA by TAp63. This should be addressed in the Discussion.

We agreed with the reviewer that the relationship between XPA expression in the different BC grades and stages and p63 expression is not as straightforward as in established cell lines. The causes are unclear. It is possible other factors may regulate XPA expression. Since we have examined only few cases, the relationship between XPA and p63, more examples are needed to further establish the relationship between XPA and p63 in human BC tumor. We addressed this issue in the Discussion in the revised manuscript as reviewer suggested.

7) Please discuss and clarify the virtual absence of both NER and BER in the NMIBC cells. Is it because these cells are sickly and difficult to culture? These tumors should be extremely sensitive to DNA damaging agent, such as ionizing radiation.

NIMBC are proficient in both NER and BER. However, human bladder mucosa cells which are highly differentiated (95% of the expression of these cells are uroplakins) are deficient in both NER and BER (Supplementary S-Figure 4). We found that no DNA repair genes are expressed in these cells (Supplementary S-Figure 4). We have published these results (Lee et al., Oncotarget, 2015). However, in the established NMIBC and MIBC cells, which are derived from BC, and uroplakins are not expressed (Lee et al., Oncotarget, 2015). We found that while MIBC cells are deficient in NER and BER, NMIBC are proficient in both. Since MIBC cells are deficient in BER, we expect that these cells are sensitive to ionizing radiation as review suggested. However, determining the ionizing radiation induced cytotoxicity, which requires ionizing radiation sources is beyond the scope of this research.

8) The high levels of unmutated (WT) p53 expression in NMIBC cells should lead to cell cycle arrest and possibly apoptosis unless a downstream protein is mutated or altered in expression. It is requested that the authors test whether the p53 damage response pathway is functional in these cells.

We found that the level of p53 expression in NMIBC is similar to, but not higher than the p53 expression in the primary cultured normal human bladder urothelial progenitors (HBEP), both carry wilt type of p53 gene (Figure 2A). We added this information in the revised manuscript (Figure 2A). These results, led us to assume that there is no a priori reason that p53 in the NMIBC would induce different p53 damage response, such as apoptosis and cell cycle effect, from normal bladder urothelial cells. In other words, we believe p53 gene function “normally” in NMIBC the same as in normal bladder urothelial cells. Therefore, we did not investigate whether or not the p53 in NMIBC cells (wild type with normal expression level as in primary cultured bladder urothelial cells) induces unusual DNA damage responses.

9) Please discuss the potential role of low levels p63 in MIBC in double strand break repair.

The repair of DNA double strand break (DSB) mainly occurs via homologous DNA recombination and non-homologous end-to-end joint. The recognition and the process of DSB repair are independent from NER and BER, thus beyond the scope of our current focus. Since we do not have any relevant results on this aspect, we believe it would be highly speculative to discuss the role of p63 in DSB repair in this manuscript. However, we believe it is a distinct possibility that p63 may regulate genes that are involved in DSB repair.

10) The authors state that the NMIBC cells are hyperactive for DNA repair but they seem to be normal. Please tone down this conclusion or clarify it with data or examples from the literature.

In the revised manuscript we present results which show that DNA repair capacity in NMIBC (RT4) is significantly higher than in primary cultured normal human bladder urothelial progenitor (HBEP) cells (100% vs 29-38%) in Figure 1E and 1F.

11) Please explain why the two wavelengths of UV are used in the two assay in Figure 1.

Low mercury pressure germicidal lamp which emits 254 nm UVC (>95%) is the only UV source used for irradiation in all experiments. We added this statement in the revised manuscript.

Reviewer #1:Figure 2A shows that all types of p63 isoforms are largely disappeared in invasive bladder cancer lines, indicating the down regulation of both TA and δ-N forms. It is suggested to examine the regulation of DNA repair by δ-N forms in these cells.

There are two major reasons for examining the roles of TAp63 isoforms but not ΔNp63 isoforms in regulating DNA repair and cell invasion function in bladder cancer cells. First, we have found that no p63 expression, including all the TAp63 and ΔNp63 isoforms, at either the protein and mRNA level, were detected in the muscle invasive bladder cancer (MIBC) cells such as T24, HT1197 and J82 cells (Figure 2 and Supplementary S-Figure 2). On the other hand, the TAp isoforms are highly expressed in non-invasive bladder cells (NMIBC) RT4 at both mRNA and protein level (Figure 2 and Supplementary S-Figure 2). At protein level ΔNp63 isoforms were barely or not detectable in the NMIBC RT4 cells (Figure 2 and Supplementary S-Figure 2). It has been established that TAp63 isoforms have transcription activation function while ΔNp63 isoforms lack this function. Since we found that MIBC cells are deficient in NER and BER and have a suppressed expression of not only multiple DNA repair genes, such as XPC, hOGG1/2 and Ref1, but also TAP63γ. In contrast, both DNA repair genes and TAp63γ are highly expressed in NMIBC cells. These results raise the possibility the expression of these multiple repair genes requires transcription activation of TAp63γ. Second, it has been found that ΔNp63 isoforms do not function as transcription activation factor (lack of transcription activation domain) (Petitjean et al., carcinogenesis, 2008). It has been suggested that DNp63 isoforms function as inhibitors for TAp63 isoforms (via polymerization with monomeric TAp63?). Since at protein level ΔNp63 isoforms were not detectable in the MIBC, and were barely detectable or not detectable in NMIBC, these results strongly suggest that the deficient of DNA repair of MIBC has nothing to do with ΔNp63 function, and ΔNp63 isoforms may not play important role in both DNA repair and tumor invasion function.

Our results confirm that both of these reasons are correct. Our results are both necessary and sufficient to demonstrate that TAp63g role in DNA repair and invasion. These are the reasons we focused our research on determining the function of TAp63. We believe the same rationales are shared by many researchers in this area. For example, Liu et al., found that TAp63γ upregulates NER (DNA repair, 2012).

There are more than six TAp63 isoforms and six ΔNp63 isoforms have been identified in human cells. The results that MIBC cells are lacking expression of TAp63g and multiple DNA repair genes, and that while NMIBC cells lack ΔNp63 expression, they have abundant expression of TAp63γ and DNA repair genes, and that TAp63 isoforms have transcription activation function while ΔNp63 isoforms do not has led us to make a reasonable practical decision to test the role of major TAp63 isoforms including TAp63g rather than testing DNp63 isoforms in DNA repair function.

Authors examined the effect of ectopic TAp63γ expression in invasive bladder cancer only (not in non-invasive bladder cancer) and TAp63 knockdown in non-invasive bladder cancer only (not in invasive bladder cancer). It is suggested to perform these experiments in both type of cells to compare whether there are difference between invasive and non-invasive bladder cancer cells.

Our results show that MIBC cells are deficient in TAp63γ expression and repair, in contrast, NMIBC cells are repair proficient and have normal to hyper-expression of TAp63γ. To further find out the causative role of the expression of TAP63γ and the repair capacity in these two types of cells, we determined whether or not changing the expression of this isoform in MIBC and NMIBC would result in changing the repair capacity and invasion phenotypes. Our results in Figures 2-4 demonstrate that it is indeed the case. Namely, the TAp63γ expression is clearly related to the DNA repair capacity and invasion phenotypes of NMIBC and MIBC. The biochemistry results also clearly support the idea that TAp63γ regulates the DNA repair function (Figure 3). We believe these results are necessary and sufficient to support the hypothesis that TAp63γ regulates both repair and invasion function.

Since MIBC cells are deficient in TAp63γ already, it is difficult to envision that further knocking down this gene expression (to reduce p63 expression in the cells which y have zero p63 expression) will provide more information on the role of this gene in DNA repair and invasion. The same is true in the case of NMIBC, since TAp63γ is highly expressed in these cells and these cells are repair proficient and non-invasive, it is difficult to envision further forced TAp63γ expression would yield more information on the role repair and invasion. Therefore, we believe that further knocking down the TAP63γ in cells which are deficient in the expression of this gene, and forced expression of this gene in cells which are already proficient in its expression, will not render more information on the role of this protein in DNA repair and invasion function.

Figure 5 shows that in bladder cancer cell lines tested, only selective mutant p53 proteins exhibit the promoting effects on migration and invasion. The role of mutant p53 in migration and invasion in bladder cancer needs to be further characterized. Since the main topic of this study is the repair capacity in bladder cancer cell lines and the connection between p63 and DNA repair in bladder cancer cells, authors may consider remove this part of the study.

We accepted the reviewer’s critique and suggestion. We removed the results of the selective mutant p53 proteins the promoting effects on migration and invasion and revised the text accordingly.

Please provide quantitation for migration, invasion, and wound healing experiments to increase the clarity, and provide error bars for the quantitative results.

We added the quantification results with error bars on migration in addition to the invasion results in Figure 4 in the revised manuscript. The wound healing results are to show the effect of TAp63g on the migration ability of MIBC using different approach. We believe this method at best is semiquantitative. Furthermore, the results are consistent with the results obtained from chamber migration method in which we have presented the proper quantified results.

Reviewer #2:In Figure 2A, the weak exposure of p63 (4A4) blot should be presented, in order to observe the expression difference of p63 between HBEP and RT4 cells.

We added the lighter exposure picture in Figure 2A as reviewer suggested in the revised manuscript.

In Figure 3A, 3B and 3D, the authors should remove the XPA blots, otherwise, the authors would like to discuss in the main text.

Our results show that XPA is expressed in the established NMIBC (RT4) and MIBC (T24 and HT1197) cells. Forced TAp63γ expression in MIBC cells does not affect the XPA expression and knockdown TAp63γ also does not affect XPA expression in NMIBC cells. These results indicate that XPA expression is not regulated by TAp63γ. Similar results were also reported by Liu et al., (DNA Repair, 2012). However, our results show that the levels of both XPA and TAp63 in bladder tumors isolated from T3/T4 patients are relatively low compared to those in T1/T2 and Ta patients. To establish the relationship between the XPA expression and the stage of bladder cancer we need to examine a lot more samples from more T3/T4 patients. We discussed this aspect in the revised manuscript (also suggested by Reviewer #3).

I suggest to remove the Figure 4A and Figure 5A. This study lacks mechanical analyses (DNA repair, gene expression etc.) to support the idea that mutant p53 plays a role here. In addition, the data from cell lines used in this study cannot support the main statement for these two figures.

We accepted reviewer’s critique and suggestion. We removed the results of the selective mutant p53 proteins the promoting effects on migration and invasion and revised the text accordingly.

Reviewer #3:There are however three points which should be addressed in the Discussion: the virtual absence of both NER and BER in the NMIBC cells bothers me. I would expect these cells to be sickly and difficult to culture. In addition, these tumors should be extremely sensitive to DNA damaging agent, such as ionizing radiation. Secondly, the high levels of unmutated p53 expression in NMIBC cells should lead to cell cycle arrest and possibly apoptosis unless a downstream protein is mutated or underexpressed. It would have been good to know if the p53 damage response pathway was functional in these cells. However, this information is not essential to support the conclusions of this study.Finally, many cancer cell types have defects in double strand break repair. The low levels of p53 in MIBC cells should affect DSB repair. The authors could discuss this.

NMIBC cells are NER and BER proficient. I believe the reviewer meant that MIBC cells are deficient in NER and BER. We agree with the reviewer that MIBC cells should be more sensitive than NMIBC to DNA damaging agents and discuss the possibility of using DNA damaging agents for therapeutic purpose for MIBC in the revised manuscript.

It is well established that many genes, including NER and BER genes and p53, are part of DNA damage response network and that the response is DNA damaging agent dependent. We agreed with the reviewer that the information obtained from studying the DNA damaging response in MIBC versus NMIBC is not essential to support the conclusions of this study. Based on the pleiotropic functions of p53 we agreed with reviewer that MIBC and NMIBC may have different repair capacity for DSB. However, we do not have any relevant results on this aspect, we believe it would be highly speculative to discuss the role of p63 in DSB repair in this manuscript.